| Clinical Microbiology | Research Article

# A prospective study of nanopore-targeted sequencing in the diagnosis of central nervous system infections

Yu Fu,[1] Jihong Gu,[1] Liang-Jun Chen,[1] Mengyuan Xiong,[1] Jin Zhao,[1] Xiao Xiao,[1] Junying Zhou,[1] Zhiqiang Li,[2] Yirong Li[1,3,4]

**ABSTRACT** Central nervous system (CNS) infections are a leading cause of death in patients. Nanopore-targeted sequencing (NTS) has begun to be used for pathogenic microbial detection. This study aims to evaluate the ability of NTS in the detection of pathogens in cerebrospinal fluid (CSF) through a prospective study. Fifty CSF specimens collected from 50 patients with suspected CNS infections went through three methods including NTS, metagenomic next-generation sequencing (mNGS), and microbial culture in parallel. When there was an inconsistency between NTS results and the results of the mNGS, the 16S rDNA gene was amplified followed by Sanger sequencing to further verify pathogens detected by NTS. Among 50 CSF specimens, 76% were NTS-positive, which is lower than mNGS (94.0%), yet higher than microbial culture (16.0%). The overall validation rate, diagnostic accordance rate (DAR), sensitivity, specificity, positive predictive value (PPV), and negative predictive value (NPV) of NTS were 86.7%, 50.0%, 71.0%, 15.8%, 57.9%, and 25.0%, respectively. In the CSF total nucleated cell (TNC) number ≤10 cells/µL, DAR, specificity, and PPV were 20%, 11.1%, and 11.1%, whereas in that with CSF TNC number >10 cells/µL, DAR, sensitivity, specificity, PPV, and NPV were 57.5%, 70.0%, 20.0%, 72.4%, and 18.2%, respectively. Although NTS has a higher microbial detection rate than microbial culture, it should combine CSF TNC result to evaluate the value of NTS for the diagnosis of CNS infections.

**IMPORTANCE** This study aims to prospectively evaluate the ability of nanopore-targeted sequencing (NTS) in the detection of pathogens in cerebrospinal fluid (CSF). It was the first time combining mNGS and microbial culture to verify the NTS-positive results also using 16S rDNA amplification with Sanger sequencing. Although microbial culture was thought to be the gold standard for pathogens detection and diagnosis of infectious diseases, this study suggested that microbial culture of CSF is not the most appropriate way for diagnosing central nervous system (CNS) infection. NTS should be recommended to be used in CSF for diagnosing CNS infection. When evaluating the value of NTS for diagnosis of CNS infections, the results of CSF TNC should be combined, and NTS-positive result is observed to be more reliable in patients with CSF TNC level >10 cells/µL.

**KEYWORDS** nanopore-targeted sequencing, central nervous system infections, clinical indicators

Central nervous system (CNS) infections are a group of neurological diseases, with varying symptoms according to the type of infection, the causative agent, and the location where they occur (1). However, due to the overlapping and nonspecific symptoms of CNS infections at early stages, the definitive etiologic diagnosis is difficult to make, leading to high mortality and morbidity (2, 3). Inflammatory markers such as procalcitonin (PCT), interleukin-6 (IL-6), and the total white cell count, covering absolute neutrophil cell count in the periphery, would be elevated in infected patients. These markers are used as indicators of infectious diseases but are not specific for

Address correspondence to Zhiqiang Li, lizhiqiang@whu.edu.cn, or Yirong Li, liyirong838@163.com.

Yu Fu and Jihong Gu contributed equally to this article. Author order was determined by drawing straws.

The authors declare no conflict of interest.

10.1128/spectrum.03317-23 **1**

CNS infections (4, 5, 6, 7). Cerebrospinal fluid (CSF) could reflect the pathology of the nervous system. Therefore, CSF analysis, including CSF routine tests (CSF RT) (pressure, appearance, transparency, cell counts, etc.) and biochemical tests (glucose, protein, chloride, etc.), is also crucial for the diagnosis of neurological diseases, but these CSF tests fail to diagnose CNS infections at early stages. A number of diagnostic techniques are currently available for neuroinvasive pathogens ranging from culture and polymerase chain reaction (PCR) to serology (8). For instance, CSF culture is undoubtedly the gold standard for CNS infections, which can advice on antimicrobial treatment (9); nevertheless, the positive rate of CSF culture in suspected cases is significantly low (10). Positive results for pathogen-specific antigen/antibodies in serum prove the presence of infection but do not directly indicate CNS infections. PCR can now be applied for the diagnosis of bacterial and fungal CNS infections, which decreases hospitalization and duration of antibiotic therapy, including single PCR reaction or multiplex PCR with several targets which is relatively low throughput (11, 12). In addition, sequencing techniques may overcome the limitations of traditional microbiological culture and provide the opportunity for rapid reporting and improvement of infected patients' management (13). These sequencing technologies could be divided into two categories based on platforms used: (i) short-read sequencing is mainly used for metagenomic sequencing and target-amplicon sequencing (14); (ii) long-read sequencing is known for utilizing nanopore technology that could gather data about pathogens by identifying bases of DNA/RNA while they transit through a nanopore (11). Metagenomic next generation sequencing (mNGS), which is known for high-throughput and high cost, can cover nearly all pathogens but has high false positive rate because it is easily contaminated by environmental species (15). By contrast, Nanopore-targeted sequencing (NTS) is a newly developed long-read sequencing technology (16); it could overcome the limitations of both PCR and mNGS by combining the long read length (>5,000 bp) with targeted amplification identifying 16S RNA gene for bacteria and internal transcribed spacer regions 1 and 2 (ITS1/2) for fungi. But, in our previously retrospective study, it is observed that the validation rate using nested polymerase chain reaction (nested PCR) followed by Sanger sequencing was low (58.8%) in CSF specimens (17, 18). NTS assay was reported to be more sensitive than the PCR assay (19). In this prospective study, we used both CSF culture and mNGS to validate NTS results. When NTS results were inconsistent with the mNGS results, 16S rDNA amplification was then carried out followed by Sanger sequencing to validate the actual existence of pathogens in CSF samples. We also investigated the diagnostic efficiency of NTS assay for diagnosis of CNS infections prospectively.

## MATERIALS AND METHODS

### Study population

In the present study, 50 patients with suspected CNS infections admitted to the Department of Neurosurgery, Zhongnan Hospital of Wuhan University, from July 2021 to December 2021, were included in this prospective study. Inclusion criteria for patients are as follows: (i) patients were suspected of having CNS infections with typical symptoms (headache, fever, consciousness disorders, meningeal irritation, limb weakness, or vomiting); (ii) clinicians decided that CSF microbial testing and other laboratory examinations were needed; (iii) patients without contraindications of lumbar puncture including an intracranial space-occupying lesion with mass effect, posterior fossa mass, abnormal intracranial pressure, local skin infections at the lumbar puncture site, etc. (20); (iv) each CSF specimen was subjected to culture for the identification of bacteria and fungi simultaneously. Exclusion criteria are listed as follows: (i) patients with contraindications of lumber puncture as listed above; (ii) patients with clear diagnosis of noninfectious disease; (iii) patients with incomplete clinical history. General clinical information (including age, gender, and results of laboratory tests) of all included subjects was obtained by searching the electronic medical record system at Zhongnan

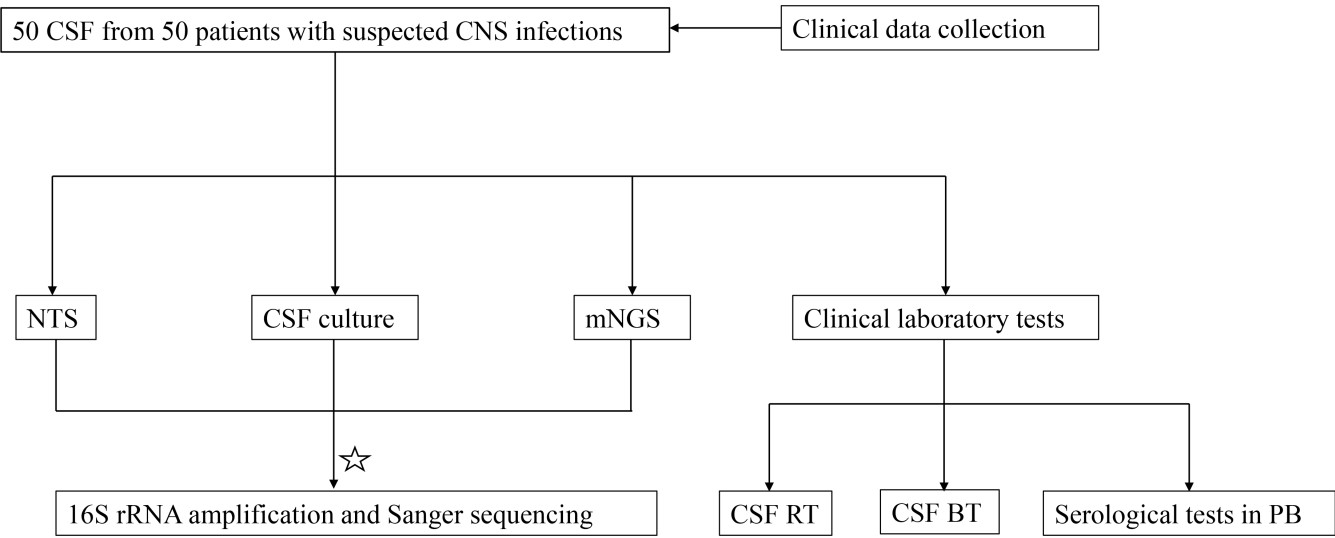

**FIG 1** The general workflow of this study. CSF BT, cerebrospinal fluid biochemical tests; PB, peripheral blood; ☆, the results of NTS did not match with the results of mNGS.

Hospital of Wuhan University. The final diagnoses of the 50 patients in this study were made upon discharge based on the comprehensive consideration of patients' clinical manifestations, laboratory tests results, and patient's clinical response to antibiotic treatment. The whole workflow of this prospective study is shown in Fig. 1.

## Specimen collection and laboratory tests

CSF specimens obtained following the standard procedures were simultaneously subjected to culture for the identification of bacteria and fungi, CSF routine, and biochemical analysis. Furthermore, 5 mL of CSF from each patient was also collected in a sterile tube and stored in a refrigerator at −80℃ for NTS and mNGS analyses. In addition, routine peripheral venous blood was collected on the day of lumbar puncture, and interleukin-6 (IL-6) and PCT levels were measured, together with white blood count (WBC) and neutrophil count (NEC).

## CSF sample preparation and nucleic acid extraction

CSF samples were centrifuged at 20,000 × $g$ for 10 min. Two hundred microliters of the sediment was kept for DNA extraction after removing the supernatant. Then, DNA was extracted from the sediment using a Nucleic Acid Extraction kit (Zybio, China) on the EXM3000 instrument (Zybio, China) following the instructions given by the manufacturer.

## Nanopore-targeted sequencing

There are several steps in the NTS analysis, which was described in our previous study (18). First, bacterial 16S rDNA gene and fungal internal transcribed spacer regions 1 and 2 (ITS1/2) were amplified using corresponding universal primers (21). Then, amplification products were purified and mixed to be used to construct sequencing libraries. Clinical samples and two Tris-EDTA buffers (no-template control, NTC) were batched in one sequencing library. Next, the library was sequenced using Oxford Nanopore GridION X5 with real-time basecalling enabled (ont-guppy-for-gridion v.1.4.3-1 and v.3.0.3-1; high-accuracy basecalling mode) (22). After that, the sequencing data were processed to discard low-quality (Q score <7) and undesired length (<200 nt or >2,000 nt). Additionally, an in-house script was used to analyze the output of the basecalling data and generate a real-time taxonomy list of each sample by screening and starting the bioinformatic pipeline when every 4,000 reads passed the base calling process; then,

**TABLE 1** Universal primers for 16S rDNA gene amplification

| Primer name | Sequence (5′–3′) | Source |
|---|---|---|
| 16S_27F | AGAGTTTGATCCTGGCTCAG | (23) |
| 16S_1429R | GGTTACCTTGTTACGACTT | |

the taxonomy of each read was assigned according to the taxonomic information of the mapped subject sequence. Finally, the interpretation of pathogen detection in the CSF sample was conducted according to a strict set of rules described elsewhere (21).

## Verification method

### Microbial culture

The CSF sample was collected and inoculated into chocolate and blood agar plates and incubated at 35°C for 12–72 h. The final identification and characterization of pathogens was done by mass spectrometry (MS) (VITEK MS system, bioMerieux, France).

### Metagenomic next-generation sequencing

In this study, the nucleic acid of CSF specimens was subjected to library preparation using the NGS Automatic Library Preparation System (Cat. MD013, Hangzhou Matridx Biotechnology Co., Ltd.; Cat. MAR002, Hangzhou Matridx Biotechnology Co., Ltd.). To control the contamination, each sequencing run on the Illumina Next Seq instrument included negative control with "no template." A bioanalyzer (Agilent 2100, Agilent Technologies, Santa Clara, CA, USA) was used due to quality management which combined with quantitative PCR before sequencing. Libraries were sequenced utilizing the Illumina NextSeq500 system with a 75-cycle sequencing kit. After removing the human-derived and low-quality data, the sequencing data were compared with the microbial nucleic acid database, and the successfully paired high-quality sequences should be further obtained from the duplicate sequences. Through further analysis and processing of the detected high-quality sequences, the pathogens present in the specimens were finally reported. This information included information such as the number of reads and coverage, aiming to assist clinical judgment of patient infection. The whole mNGS analysis was done by Hangzhou Matridx Biotechnology Co., Ltd.

### 16S rDNA amplification and sequencing

When the results of NTS did not match the outcomes of mNGS, 16S rDNA gene was amplified. The primers used for 16S rDNA amplification are shown in Table 1. A 25-μL amplification reaction system was used, including 12.5 μL of 2× Phanta Max Master Mix, 5.5 μL of ddH2O, 2 μL of primers (1 μL each of upstream and downstream primers), and 5 μL of DNA template. The amplification conditions were pre-denaturation at 94°C for 3 min, followed by denaturation at 94°C for 30 s, annealing at 58°C for 30 s, extension at 72°C for 15 s, for a total of 35 cycles, and finally extension again at 72°C for 5 min.

## Statistical analysis

Data were analyzed by IBM SPSS Statistics 26.0. The $\chi$ test was used to compare the diagnostic performance of NTS and culture for CNS infectious diseases and the positive rate of the three detection methods. Unpaired $t$ test was used to compare clinical laboratory indicators between the CNS infection group and non-CNS infection group. $P$ values less than 0.05 were considered statistically significant.

## RESULTS

### Demographic and clinical data

Fifty hospitalized patients included in this study were numbered consecutively, and the details are shown in Table 2. Twenty-five males and 25 females, with a median age

**TABLE 2** Demographic and clinical data

| Characteristics | Value | Percentage |
| --- | --- | --- |
| Median age | 57 | |
| Age group | | |
| ≤20 | 3 | 6.0% |
| 21–40 | 7 | 14.0% |
| 41–50 | 7 | 14.0% |
| 61–80 | 14 | 28.0% |
| 51–61 | 18 | 36.0% |
| >80 | 1 | 2.0% |
| Gender | | |
| Male | 25 | 50% |
| Female | 25 | 50% |
| Clinical diagnosis | | |
| CNS infections | 31 | 62.0% |
| Non-CNS infections | 19 | 38.0% |
| Culture positive | 8 | 16.0% |
| Culture negative | 42 | 84.0% |
| Clinical symptoms | | |
| Fever | 34 | 68.0% |
| Consciousness disorders | 25 | 50.0% |
| Headache | 19 | 38.0% |
| Meningeal irritation | 18 | 36.0% |
| Limb weakness | 16 | 32.0% |
| Vomiting | 15 | 30.0% |
| Mental and behavior disorders | 3 | 6.0% |
| Seizures | 1 | 2.0% |
| Clinical outcome | | |
| Cured | 41 | 82.0% |
| Self-discharge | 9 | 18.0% |

of 57 years, were all suspected of having CNS infections. The clinical manifestations of these patients were mostly fever and impaired consciousness, whereas headache, nausea, and vomiting, positive signs of meningeal inflammation, mental and behavioral abnormalities were found in minority cases. Eighty-two percent of these patients were cured in hospital. Based on the clinical diagnosis on discharge, these patients could be divided into two categories: (i) CNS infections (31/50, 62.0%) and (ii) non-CNS infections (19/50, 38.0%).

## Clinical laboratory indicators

Peripheral venous blood of each patient was collected and examined on the day of admission. Serological indicators in peripheral blood of these patients are listed in Table S1. The results of CSF tests including total nucleated cell (TNC), protein, glucose, chloride, lactate dehydrogenase (LDH), and lactate are presented in Table S2. The statistical significance of these clinical indicators is presented in Table 3.

## The testing results by NTS, mNGS, and culture

Among 50 CSF specimens, 38 (76%) were NTS-positive, and the detailed results are given in Table 4. Thirty-seven bacterial species with a total of 60 strains were detected by NTS. The bacteria with the highest detection rate were *Staphylococcus aureus* (6/60, 10.0%), followed by Pseudomonas stutzeri (4/60, 6.7%), *Pseudomonas aeruginosa* (3/60, 5.0%), *Klebsiella pneumoniae* (3/60, 5.0%), and *Staphylococcus epidermidis* (3/60, 5.0%). The positive rate of mNGS in 50 CSF samples was 94.0% (47/50). A total of 88 bacterial strains belonging to 27 species were detected by mNGS. mNGS detected the highest rate of

**TABLE 3** Clinical laboratory indicators between CNS infection group and Non-CNS infection group[a]

| Clinical indicator | CNS infection group (n = 31) | Non-CNS infection group (n = 19) | Mean difference ± SEM | Unpaired t test P-value |
|---|---|---|---|---|
| TNC (per mm$^3$) | 1884 | 173.4 | 1711 ± 775.9 | 0.0323[b] |
| CSF-TP (g/L) | 2.903 | 1.358 | 1.545 ± 0.8383 | 0.0715 |
| CSF-Glu (mmol/L) | 1.785 | 4.249 | −2.465 ± 0.3373 | <0.0001[b] |
| CSF-Cl (mmol/L) | 100.4 | 127.2 | −26.75 ± 10.41 | 0.0133[b] |
| CSF-LDH (U/L) | 396.2 | 230.5 | 165.7 ± 150.3 | 0.2759 |
| CSF LA (mmol/L) | 60.25 | 3.609 | 56.64 ± 27.32 | 0.0468[b] |
| PCT (ng/mL) | 2.26 | 0.39 | 1.870 ± 0.8163 | 0.0308[b] |
| IL-6 (pg/mL) | 60.35 | 76.32 | −16.43 ± 37.48 | 0.6632 |
| WBC (x10 $^9$/L) | 13.37 | 11.35 | 2.013 ± 1.426 | 0.1645 |
| NEC (x10 $^9$/L) | 11.39 | 9.358 | 2.028 ± 1.371 | 0.1458 |

[a]TNC, total nucleated cell; TP, total protein; Glu, glucose; Cl, chloride; LDH, lactate dehydrogenase; LA, lactate; PCT, procalcitonin; IL-6, interleukin-6; WBC, white blood count; NEC, neutrophil count.
[b]$P < 0.05$, statistically significant.

*Burkholderia cepacia* (22/88, 25.0%), followed by *S. aureus* (18/88, 20.5%), *K. pneumoniae* (9/88, 10.2%), and *Escherichia coli* (7/88, 8.0%), respectively. The positive rate of culture in 50 CSF samples was 16.0% (8/50), and the detailed results are also given in Table 4. A total of nine isolates were cultured in eight CSF samples, including two *Acinetobacter johnsonii*, two *K. pneumoniae*, one *Staphylococcus hominis,* one Streptococcus agalactiae, one *P. aeruginosa*, one *Serratia marcescens,* and one strain of *S. epidermidis*. Of them, *S. marcescens* and *Actinobacillus jovis* were cocultured in case no. 29. The total positive rate of NTS in pathogen detection of CSF specimens was obviously lower than that of mNGS (*P* = 0.02) and yet significantly higher than that of traditional culture (*P* < 0.01).

## Verification of NTS-positive results

Of the 60 strains detected by NTS, 42 (70.0%) were able to be identified simultaneously by mNGS, as given in Table 4. Among 42 strains, 7 (16.7%) were culture positive, including 2 *K. pneumoniae*, 1 S.treptococcus agalactiae, 1 *P. aeruginosa*, 1 *S. marcescens*, 1 strain of *S. hominis*, and 1 strain of *A. johnsonii* (Table 4). Because of inconsistent results between NTS and mNGS/culture, 18 strains were detected by NTS in the 11 specimens that were further verified by 16S rDNA amplification followed by Sanger sequencing. Of them, 10 strains were able to be successfully verified by 16S rDNA amplification, including 2 for each of *S. epidermidis, Streptococcus mitis* and *Pseudomonas stutzeri*, and 1 for each of *Staphylococcus aureus, Staphylococcus warneri, Streptococcus oralis,* and *Enterobacter cloacae* (Table 4). Therefore, the total verification rate of NTS-positive results was 86.7% (52/60).

## The diagnostic performance of NTS compared with culture results

Among the 50 CSF specimens, 8 (16.0%) were culture positive and all were confirmed with CNS infections, as given in Table 4, but only 7 out of 9 cultured strains (77.8%) could be determined by NTS. In CSF specimen no. 29, the culture result was *S. marcescens* and *A. jovis*; whereas NTS detected only *S. marcescens*, and in no. 23, the culture result was only *S. epidermidis*, yet NTS detected to be *Pseudomonas luteola*, *E. coli*, *Acidovorax facilis,* and *Acinetobacter junii*. Diagnostic performance of NTS and culture for CNS infectious disease are listed in Table 5. It was showed that NTS approach had a higher diagnostic sensitivity than culture (71.0% vs 25.8%, *P* < 0.01). In 31 patients with confirmed CNS infections, 23 patients had negative culture results, and NTS was able to assist in confirming 14 (60.9%) patients with culture-negative CNS infections. But, the specificity and positive predictive value (PPV) of culture were higher than those of the NTS, whereas no significant difference was found in the negative predictive value (NPV) and the diagnostic accordance rate.

**TABLE 4**  Results of NTS assay and other methods*a*

| Sample no. | NTS result | Reads | Culture result | mNGS | 16S rDNA | Clinical diagnosis |
|---|---|---|---|---|---|---|
| 1 | *Aerococcus viridans* | 290 | Neg | Neg | Neg | N |
| | *Pantoea calida* | 183 | | Neg | Neg | |
| | *Streptococcus oralis* | 138 | | Neg | Neg | |
| 2 | NA | | Neg | ***Staphylococcus aureus*** | Pos | I |
| 3 | ***Pantoea calida*** | 65 | Neg | Neg | Neg | I |
| 4 | NA | | Neg | ***Staphylococcus aureus*** | Pos | I |
| 5 | *Granulicatella elegans* | 33 | Neg | Neg | Neg | N |
| | *Streptococcus mitis* | 25 | | Neg | Pos | |
| 6 | *Staphylococcus aureus* | 72 | Neg | Pos | | N |
| | *Burkholderia stabilis* | 190 | | Pos | | |
| 7 | *Actinomyces viscosus* | 47 | Neg | Neg | Neg | N |
| | *Staphylococcus epidermidis* | 30 | | Neg | Pos | |
| 8 | *Staphylococcus epidermidis* | 332 | Neg | Neg | Pos | N |
| | Sporosarcina | 209 | | Neg | Neg | |
| 9 | *Elizabethkingia anophelis* | 901 | Neg | Pos | | N |
| | *Chryseobacterium meningosepti-cum* | 1,067 | | Pos | | |
| | *Acinetobacter* sp. | 17,435 | | Pos | | |
| 10 | ***Pseudomonas stutzeri*** | 40 | Neg | Neg | Pos | I |
| | *Enterobacter cloacae* | 331 | | Neg | Pos | |
| 11 | NA | | Neg | | | N |
| 12 | *Burkholderia metallica* | 126 | Neg | Pos | | N |
| 13 | *Aerococcus viridans* | 30 | Neg | Neg | Neg | N |
| 14 | ***Serratia marcescens*** | 220 | Neg | Pos | | I |
| | ***Klebsiella pneumoniae*** | 47 | | Pos | | |
| 15 | NA | | Neg | ***Staphylococcus aureus*** | Pos | I |
| 16 | *Bacillus cereus* | 754 | Neg | Neg | Neg | N |
| | *Staphylococcus warneri* | 117 | | Neg | Pos | |
| 17 | NA | | Neg | ***Burkholderia cepacia*** | Neg | I |
| 18 | ***Streptococcus mitis*** | 12 | Neg | Neg | Pos | I |
| 19 | *Staphylococcus aureus* | 482 | Neg | Pos | | N |
| | *Propionibacterium acnes* | 166 | | Pos | | |
| | *Streptococcus oralis* | 82 | | Pos | | |
| 20 | *Pseudomonas stutzeri* | 97 | Neg | Neg | Pos | N |
| 21 | NA | | Neg | | | N |
| 22 | *Pseudomonas stutzeri* | 97 | Neg | Pos | | N |
| 23 | ***Pseudomonas luteola*** | 307 | *Staphylococcus epidermidis* | Pos | | I |
| | *Escherichia coli* | 62 | | Pos | | |
| | *Acidovorax facilis* | 1,791 | | Pos | | |
| | *Acinetobacter junii* | 211 | | Pos | | |
| 24 | *Staphylococcus aureus* | 498 | Neg | Pos | | N |
| 25 | *Staphylococcus epidermidis* | 166 | Neg | Pos | | I |
| | ***Acinetobacter junii*** | 94 | | Pos | | |
| 26 | *Pseudomonas stutzeri* | 57 | Neg | Pos | | N |
| | *Propionibacterium acnes* | 59 | | Pos | | |
| 27 | *Pseudomonas aeruginosa* | 16 | Neg | Pos | | N |
| 28 | ***Klebsiella pneumoniae*** | 8,090 | ***Klebsiella pneumoniae*** | Pos | | I |
| 29 | ***Serratia marcescens*** | 3,450 | ***Serratia marcescens*** | Pos | | I |
| | | | *Acinetobacter johnsonii* | | | |
| 30 | NA | | Neg | | | N |
| 31 | ***Staphylococcus hominis*** | 34,555 | ***Staphylococcus hominis*** | Pos | | I |
| | *Pseudomonas aeruginosa* | 5,392 | | Pos | | |

**TABLE 4** Results of NTS assay and other methods[a] (*Continued*)

| Sample no. | NTS result | Reads | Culture result | mNGS | 16S rDNA | Clinical diagnosis |
|---|---|---|---|---|---|---|
| 32 | *Acinetobacter johnsonii* | 45,265 | *Acinetobacter johnsonii* | Pos | | I |
| | *Pseudomonas fluorescens* | 9,754 | | Pos | | |
| 33 | NA | | Neg | *Klebsiella pneumoniae* | Pos | I |
| | | | | *Escherichia coli* | | |
| 34 | NA | | Neg | *Burkholderia cepacia* | | I |
| 35 | *Acinetobacter johnsonii* | 175 | Neg | Pos | | I |
| 36 | *Staphylococcus aureus* | 56 | Neg | Pos | | N |
| 37 | *Staphylococcus aureus* | 583 | Neg | Pos | | I |
| 38 | NA | | Neg | *Escherichia coli* | | I |
| 39 | *Pseudomonas aeruginosa* | 92,017 | *Pseudomonas aeruginosa* | Pos | | I |
| 40 | *Klebsiella pneumoniae* | 31,912 | *Klebsiella pneumoniae* | Pos | | I |
| | *Enterobacter cloacae* | 330 | | Pos | | |
| 41 | NA | | Neg | *Burkholderia cepacia* | | I |
| 42 | *Actinomyces Cardiff* | 3,471 | Neg | Pos | | I |
| 43 | *Staphylococcus caprae* | 720 | Neg | Pos | | I |
| 44 | NA | | Neg | *Burkholderia cepacia* | | I |
| 45 | *Achromobacter xylosoxidans* | 45 | Neg | Pos | | I |
| 46 | *Escherichia coli* | 406 | Neg | Pos | | I |
| 47 | *Streptococcus anginosus* | 37 | Neg | Pos | | I |
| | *Parvimonas micra* | 1,132 | | Pos | | |
| 48 | *Streptococcus agalactiae* | 8,585 | *Streptococcus agalactiae* | Pos | | I |
| 49 | *Staphylococcus aureus* | 188 | Neg | Neg | Pos | I |
| 50 | *Mycobacterium neworleansense* | 3,868 | Neg | Pos | | I |

[a]Bacterial names in bold highlight the identified pathogen diagnosed as CNS infections. NA, not applicable; I, CNS infections; N, non-CNS infections; neg (in mNGS), the mNGS results were not consistent with the species shown by the NTS and may detect bacteria or may not detect anything at all; pos (in mNGS), the results of mNGS were consistent with those of NTS.

## Diagnostic efficiency of NTS in CNS infections

It was found that the diagnostic accordance rate of NTS varied from 20.0% to 69.2% in different CSF TNC number groups (Table 6). In the group where CSF TNC number of ≤10 cells/μL, the diagnostic accordance rate was only 20%, with both the specificity and PPV of 11.1%. On the other hand, in the remaining three groups with TNC number in CSF of >10 cells/μL, all of the diagnostic accordance rates were ≥50%. We then combined them into one group and found the diagnostic accordance rate was 57.5%, with the sensitivity of 70.0% and the specificity of 20.0%. The PPV and NPV were 72.4% and 18.2%, respectively.

## DISCUSSION

In this prospective study, it was the first time combining mNGS, another high throughput sequencing approach, and microbial culture to verify the NTS-positive results also using 16S rDNA amplification with Sanger sequencing. Seventy percent of NTS-positive strains were verified by mNGS, and the total verification rate was 86.7%. In addition, of 38 strains determined to be less than 400 mapped reads by NTS, the verification rate was 55.3%. In our previously retrospective study, an average of 85.2% NTS-positive results in six types of clinical samples were verified successfully using PCR followed by Sanger sequencing, with a maximum of 95.8% in hydrothorax and ascites and a minimum of 58.8% in CSF, and moreover, of 7 strains tested to be less than 400 mapped reads by NTS, only 1 (14.3%) strain was verified (18). It was reasonable to speculate that the use of mNGS approach may improve the verification rate for NTS-positive strain. Both mNGS and NTS approaches have higher values of sensitivity than that of the PCR assay. In line with our speculation, a previous study found that the NTS assay could be approximately 100 times more sensitive than the real-time PCR assay (19).Therefore, in samples with low bacterial

**TABLE 5** Diagnostic performance of NTS and culture for CNS infectious diseases[a]

|  | CNS infection | Non-CNS infection | SN | SP | PPV | NPV | DAR |
|---|---|---|---|---|---|---|---|
| NTS positive | 22 | 16 | 71.0% | 15.8% | 57.9% | 25.0% | 50.0% |
| NTS negative | 9 | 3 |  |  |  |  |  |
| Culture positive | 8 | 0 | 25.8% | 100.0% | 100.0% | 45.2% | 54.0% |
| Culture negative | 23 | 19 |  |  |  |  |  |
| P |  |  | <0.01 | <0.01 | 0.03 | 0.32 | 0.42 |

[a]SN, sensitivity; SP, specificity; PPV, positive predictive value; NPV, negative predictive value. DAR, diagnostic accordance rate. The diagnostic coincidence rate means the consistency between the diagnosis based on the test results of a method (NTS) and the final diagnosis results (based on the typical clinical symptoms of CNS infections and microbial culture, etc.).

**TABLE 6** Diagnostic accordance rate by NTS varied in different TNC groups[a]

| TNC (cells/μL) | Number of samples | NTS positive | | NTS negative | | DAR (95% CI) |
|---|---|---|---|---|---|---|
|  |  | Infected | Noninfected | Infected | Noninfected |  |
| ≤10 | 10 | 1 (10.0%) | 8 (80.0%) | 0 | 1 (10.0%) | 20.0% (−0.102–0.502) |
| 10 < n ≤ 100 | 13 | 6 (46.2%) | 3 (23.1%) | 3 (23.1%) | 1 (7.7%) | 53.8% (0.225–0.852) |
| 100 < n ≤ 1000 | 14 | 6 (42.9%) | 5 (35.7%) | 2 (14.3%) | 1 (7.1%) | 50.0% (0.2–0.8) |
| >1000 | 13 | 9 (69.2%) | 0 | 4 (30.8%) | 0 | 69.2% (0.402–0.983) |

[a]DAR, diagnostic accordance rate. 95% CI, confidence intervals.

burden, including CSF and blood, it is difficult to verify the NTS-positive result only using a PCR assay.

Microbial culture was thought to be the gold standard for pathogens detection and diagnosis of infectious diseases (24). In the present study, its sensitivity and PPV for diagnosing CNS infection was 25.8% and 100.0%, respectively, which indicates that microbial culture of CSF is not the most appropriate way for diagnosing CNS infection. Although the NTS assay has the same diagnostic accordance rate as microbial culture, the sensitivity of NTS assay was obviously higher than that of microbial culture, and moreover, NTS was able to assist in confirming 60.9% patients with culture-negative CNS infections in this present study. Due to the strong capacity of pathogen detection, NTS should be recommended to be used in CSF for diagnosing CNS infection. Supporting our view, a pilot study including six retrospective cases of confirmed bacterial meningitis and two prospective cases showed that the NTS was more sensitive than conventional culture and worked properly even in samples exposed to antibiotics (25). Another study showed that the diagnostic sensitivity of the NTS and culture was 100% and 44.9%, respectively, for the genuine CNS infection (26). Although the NTS assay was found to have a relatively high sensitivity of 71.0% for diagnosing CNS infections, it presented positive results in 16 of 19 patients diagnosed with non-CNS infections, which caused a relatively low PPV and a relatively high false positive rate. There are several reasons for positive NTS results in noninfected patients. For instance, CSF samples contamination during sample collection and transportation, reagents carrying a minority of DNA from engineering bacteria and environmental bacterial contamination during the NTS assay (27, 28). Although these DNA present are very low levels, it can be more easily amplified in DNA solution with very low bacterial DNA burdens, especially in that with very low DNA burdens, which attributed to the deficiency of competitive or noncompetitive inhibition during PCR reaction (29). Therefore, the interpretation of NTS positive results needs to be closely related to the clinical manifestations of CNS infection (fever, headache, nausea, coma, neck tonicity, etc.) to exclude normal microbial colonization as well as contamination.

In the present study, when CSF TNC number was less than 10 cells/μL, the diagnostic accordance rate was only 20% with both the specificity and PPV of 11.1%, whereas TNC number in CSF >10 cells/μL, the diagnostic accordance rate was 57.5% with the sensitivity of 70.0% and the specificity of 20.0%; and the PPV and NPV were 72.4% and 18.2%. Because of low PPV, NTS is not suggested to be used for the detection of pathogens in patients with suspected CNS infections under the condition of TNC number

≤10 cells/µL. Once the TNC number in CSF of >10 cells/µL, the PPV has increased to 72.4% with low NPV of 18.2%, indicating that NTS-positive result is observed to be more reliable than NTS-negative results. There were several reasons for interpreting the low NPV levels, including PCR amplification inhibition from high levels of protein from TNCs, primer mismatching, and self-limited viral infections.

NTS has a turnaround time of about 8–12 h from sample collection to final report, which is much shorter than that of microbial culture (12–72 h) and mNGS (around two working days) (30). Therefore, NTS would be most suitable to detect causative agents in CSF from patients suspected of CNS infections, particularly in those having more than 10 cells/µL of CSF TNC.

However, our study has some disadvantages: (i) due to its targeted nucleic acid amplification for bacteria and fungi, it could not be used for diagnosis of viral CNS infections; (ii) an insufficient number of cases, the number of CSF samples with TNC less than or equal to 10 cells/µL was low; (iii) NTS could not generate data about clinical epidemiological information including strain typing, virulence, and drug-resistance genes.

## Conclusion

Compared with traditional microbiology laboratory tests, NTS can rapidly identify microorganisms in CSF specimens. However, clinicians should be cautious in interpreting the NTS-positive results, which are easily contaminated by environmental species and normal flora. It should combine CSF TNC results to evaluate the value of NTS for diagnosis of CNS infections. NTS-positive result is observed to be more reliable in patients with CSF TNC level >10 cells/µL. Moreover, NTS is not suggested to be used in patients with suspected CNS infections having CSF TNC level less than 10 cells/µL.

## ACKNOWLEDGMENTS

The authors thank all the colleagues and the reviewers who helped with this work.

This work was supported by the Hubei Province Key Research and Development Project (2022BCA019).

Yu Fu: writing original draft, data curation, and investigation process. Jihong Gu: resources, visualization. Liangjun Chen: formal analysis, validation, and conceptualization. Mengyuan Xiong, Jin Zhao: resources and software. Xiao Xiao, Junying Zhou: methodology and conceptualization. Zhiqiang Li: fund acquisition and methodology. Yirong Li: fund acquisition, review and editing.

The authors declare that the research was conducted in the absence of any commercial or financial relationships.

## AUTHOR AFFILIATIONS

[1]Department of Clinical Laboratory, Zhongnan Hospital of Wuhan University, Wuhan, China
[2]Department of Neurosurgery, Zhongnan Hospital of Wuhan University, Wuhan, China
[3]Wuhan Research Center for Infectious Diseases and Cancer, Chinese Academy of Medical Sciences, Wuhan, China
[4]Hubei Engineering Center for Infectious Disease Prevention, Control and Treatment, Wuhan, China

## AUTHOR ORCIDs

Jihong Gu  http://orcid.org/0000-0001-9906-9478
Zhiqiang Li  http://orcid.org/0000-0003-0144-8790
Yirong Li  http://orcid.org/0000-0002-5619-1614

## AUTHOR CONTRIBUTIONS

Yu Fu, Formal analysis, Investigation, Methodology, Writing – original draft | Jihong Gu, Resources, Software, Validation | Liang-Jun Chen, Conceptualization, Methodology, Validation | Mengyuan Xiong, Resources, Software | Jin Zhao, Resources, Software | Xiao Xiao, Conceptualization, Methodology | Junying Zhou, Conceptualization, Methodology | Zhiqiang Li, Funding acquisition, Methodology | Yirong Li, Funding acquisition, Writing – review and editing

## ETHICS APPROVAL

This prospective study was approved by the Medical Ethics Committee, Zhongnan Hospital of Wuhan University (2022137K). Written informed consent was obtained from all patients.

## ADDITIONAL FILES

The following material is available online.

### Supplemental Material

**Table S1 (Spectrum03317-23-S0001.docx).** Serological indicators in 50 patients with suspected CNS infections.

### Open Peer Review

**PEER REVIEW HISTORY (review-history.pdf).** An accounting of the reviewer comments and feedback.

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
