## [Reviewer comments · Microbiology Spectrum]

Microbiology Spectrum

A Prospective Study of Nanopore Targeted Sequencing in the diagnosis for Central Nervous System Infections

Yu Fu, Jihong Gu, Liang-Jun Chen, Mengyuan Xiong, Jin Zhao, Xiao Xiao, Junying Zhou, Zhiqiang Li, and Yirong Li

Corresponding Author(s): Yirong Li, Zhongnan Hospital of Wuhan University, China

Review Timeline:

Submission Date:	September 8, 2023
Editorial Decision:	October 31, 2023
Revision Received:	December 11, 2023
Editorial Decision:	December 18, 2023
Revision Received:	December 30, 2023
Accepted:	January 4, 2024

Editor: Kessendri Reddy

Reviewer(s): Disclosure of reviewer identity is with reference to reviewer comments included in decision letter(s). The following individuals involved in review of your submission have agreed to reveal their identity: Ke Lin (Reviewer #2)

Transaction Report:

DOI: <https://doi.org/10.1128/spectrum.03317-23>

Re: Spectrum03317-23 (A Prospective Study of Nanopore Targeted Sequencing in the diagnosis for Central Nervous System Infections)

Dear Dr. Yirong Li:

Thank you for the privilege of reviewing your work. Below you will find my comments, instructions from the Spectrum editorial office, and the reviewer comments.

Revision Guidelines

Sincerely,
Kessendri Reddy
Editor
Microbiology Spectrum

Additional comments:

Major comments:

- English language review needed. "Microbial culture is difficult to guide rational administration of drugs at an early stage" (lines 26-27) does not make sense. What is the diagnostic accordance rate (mentioned in the abstract under results)?
- Lines 168-169: Were there any instances in which the NTS results did not match just the culture or just the mNGS (i.e. not

both)? What was done in those instances?

- Lines 177-178: Further information about the statistical analysis should be given, i.e. categorical variables were summarised using proportions, etc. The Chi-square test is mentioned in lines 214-215 but is not mentioned in this paragraph.
- Table 3 does not show a statistical significance comparison as mentioned in lines 195-196, it just shows the ?mean ?median of these values. Please apply and report on the appropriate statistical testing for these parameters. The table should include detail on whether mean or median was reported and the standard deviations or interquartile ranges. The inclusion of a biostatistician may be useful.
- Table 3: Abbreviations should be explained in the table footnotes.
- Table 4: This is difficult to follow. Some lines do not have mNGS results, e.g. lines 33, 34, 41, 15. Please review and correct. It would be useful to highlight pathogens that are recognised CNS pathogens or that were thought to be significant in these patients. A clinician/clinical microbiologist could assist with this. Reporting this would help the reader determine the usefulness of these techniques in clinically suspected CNS infections and would exclude the background noise resulting from uncommon pathogens being detected by these highly sensitive techniques.

Minor comments:

- Line 52: Replace "non-specific symptoms" with "overlapping and non-specific symptoms"
- Line 166: Please check the spelling of "Matridx" and provide the company headquarters for the company mentioned.
- In Table 2, the first line reads "age (year), median" but only a single number is given. The caption should be more descriptive.
- Line 185: Replace "meningitis stimulation" with "meningeal inflammation"
- Line 212: Correct spelling "epidermidis".
- Tables 5 and 6: What does the diagnostic accordance rate refer to? Please provide a definition in the methods, this is not a standard term.
- In Table 5, suggest leaving out the Chi-square value and just reporting the p-value.
- Lines 234-235: Can the authors explain why specimen 23 was totally discrepant between culture and NTS? Could there have been laboratory contamination with the coagulase negative staphylococcus during culture? Could the mNGS help to clarify the discordance?
- Line 256: The sentence cannot start with a number. Please write out "Seventy percent" or reword the sentence.
- Line 283: Reword to "samples exposed to antibiotics".

Reviewer #1 (Comments for the Author):

The article titled "A Prospective Study of Nanopore Targeted Sequencing in the diagnosis for Central Nervous System Infections" by Fu, Y et. al., presents the possibility of nanopore sequencing as an alternative to diagnose neurological infections. The article highlights the existing difficulty in the diagnosis of neurological infections effectively and provides a comprehensive comparison of the conventional gold-standard methods such as culture against PCR and latest metagenomic and nanopore sequencing technologies. It shows that the gold-standard culture is not always the best method considering the variety of factors involved including the microbial load and ability of the microbe to grow in culture. The work effectively emphasizes on the need for better diagnostic tools and provide an alternative. The authors perspective and discussion on the effect of the total cellular content in the CSF is interesting. It is a commendable work that makes an important contribution to the field. However, there are a few concerns regarding the work that I feel need to be addressed before considering for the publication.

Major concerns:

1. The authors have considered 50 CSF samples from suspected neuro-infection cases and considered the ones that were clinically exempted as non-infection as negative controls. However, the manuscript failed to inform the readers the basis of this clinical outcome.
2. Though the authors have included a no template control for illumine sequencing as a control, there is no mention of a similar control for the Nanopore sequencing.
In addition, the reviewer think that additional true negative controls is necessary to substantiate the suitability of the method for diagnostic purposes. Probably a CSF samples collected from patients diagnosed with non-neuroinfectious condition would serve as better controls to calculate the false negative rate of the sequencing methods
3. It would be interesting to see how additional positive controls of a few of the microbial species, perhaps from pure cultures would perform through the same sequencing pipeline. This would provide better statistics for the use of this method as diagnostic tool.

Minor Comments:

1. It would help to specify the criteria for "low-quality reads" and the "undesired length"
2. The results of the MALDI-TOF experiments need to be included in the Table 4
3. It would help to provide the basis and the final clinical diagnosis of these patients

Reviewer #2 (Comments for the Author):

In this article, Li and colleagues conducted a diagnostic efficacy assessment of Nanopore Targeted Sequencing (NTS) for its clinical application in central nervous system (CNS) infections. They compared it with current commonly used mNGS detection methods and the gold standard culture method on cerebrospinal fluid (CSF) from 50 patients with suspected CNS infection. And CSF with disputed test results were further detected by 16S rRNA amplification and Sanger sequencing. This study represents a novel application of NTS technology in the field of CNS infections.

Comments:

1. In evaluating Nanopore's diagnostic efficacy for CNS, there may be an insufficient number of cases. In this manuscript, 50 patients with suspected CNS infection were enrolled, while only 31 of them were diagnosed with CNS infection. It could be beneficial to validate its performance for specific types of pathogens, such as assessing the sensitivity and specificity of NTS in CNS fungal infections.
2. In clinical practice, mNGS has already been used in many hospitals for detecting pathogens in CSF. The authors could further compare NTS and mNGS detection in terms of reporting time and expenses to stress the practical value of application of NTS in CNS infection.
3. Overall, the authors should further emphasize the application value and advantages of NTS in CNS infections. Additionally, it would be beneficial to include images representing the application of NTS technology in CNS infection for better understanding.

Dear Editors:

The authors' conclusions are supported by data. The language of the manuscript still needs to be polished and revised. The authors evaluated the diagnostic performance of NTS for central nervous system infections (CNSIs) among patients. While the study design limits the applicability of the findings. The small number of cases included in the study makes the results unpowerful. Moreover, the research design and the writing of the document lack rigor. Therefore, my recommendation is to reject.

Comments and Suggestions for the Author:

This study aims to comprehensively evaluate the ability of NTS in the detection of pathogens in cerebrospinal fluid (CSF) through a prospective study. Some comments are provided that may strengthen the manuscript.

Major comments:

1. The inclusion and exclusion criteria of this study were not rigorous. First, both inclusion criteria and exclusion criteria refer to "contraindications of lumbar puncture", which can be placed in the exclusion criteria. Second, the inclusion of "clinician believes that cerebrospinal fluid examination is necessary" in the criteria is a subjective criterion, please replace it with an objective and specific criterion.
2. The diagnostic criteria, which divide enrolled patients into infected and non-infected groups, should be included in the text. Therefore, please supplement the diagnostic criteria for the final diagnosis in the text, and should cite recognized references.
3. Small case number is a major concern. Only 31 cases with CNS infection were enrolled in this study which will lose enough power to yield statistically convincing results.
4. In this study, the results of mNGS and culture were used to support the results of NTS, but there are still problems that mNGS is false positive. Culture is currently the accepted gold standard for diagnosing central nervous system infections. NTS and culture inconsistencies should be validated by 16s RNA.
5. It is recommended that the sequencing data of mNGS and NTS be uploaded to a public website.
6. The discussion section of this article needs to be revised. A large number of results sections are repeated in the discussion. Please discuss the clinical significance of the study based on the findings. First, according to the research results, NTS has a high false positive rate. How to explain this problem? Second, since mNGS has a better positive detection rate than NTS, what are the advantages of using NTS? Then, the limitations of this study should also be included in the discussion.
7. Due to the small number of included cases, whether CSF TNC < 10 cells/ μ l can be used as cut off value remains to be discussed.

Minor Comments:

1. The flowchart in Figure 1 was brief, please improve the supplementary content, such as inclusion criteria and exclusion criteria.
2. Among the 30 references in this paper, 17 are from the last five years, and only 9 are from the last three years. Please add references in 3-5 years.
3. The methodological description of statistical analysis is too brief, please complete it.
4. Sensitivity and specificity are not appropriate in this context. positive percent agreement and negative percent agreement should be used when comparing with the non-gold standard.

Dear Editor

Thank you for your evaluation of our manuscript. Those suggestions are all valuable and very helpful for revising and improving our paper, as well as the important guiding significance to our researches. Following their suggestions, we have carefully revised our paper and addressed all their concerns. Revised portions are marked in revisions mode in the paper and point-by-point responses to these comments are listed below.

Best regards,

Yirong Li

Liyirong838@163.com

Response letter to Editor and Reviewers

Manuscript ID number: Spectrum03317-23

Title of paper: A Prospective Study of Nanopore Targeted Sequencing in the diagnosis for Central Nervous System Infections

Editor decision

Major comments:

- English language review needed. “Microbial culture is difficult to guide rational administration of drugs at an early stage” (lines 26-27) does not make sense. What is the diagnostic concordance rate (mentioned in the abstract under results)?

RESPONSE: Thank you for your valuable suggestions. We have deleted the “Microbial culture is difficult to guide rational administration of drugs at an early stage” accordingly”. The diagnostic coincidence rate means the consistency between the test results of a method (NTS) and the final diagnosis results (based on the typical clinical symptoms of CNS infections and microbial culture, etc.). We have added this definition to the notation in Table 5.

- Lines 168-169: Were there any instances in which the NTS results did not match just the culture or just the mNGS (i.e. not both)? What was done in those instances?

RESPONSE: I apologize for our mischaracterization. In fact, we performed 16S rDNA amplification and sequencing when any two of the three methods did not agree (Table 4 proves this). We have corrected it in the manuscript (line 100 and line 178).

Lines 177-178: Further information about the statistical analysis should be given, i.e. categorical variables were summarised using proportions, etc. The Chi-square test is mentioned in lines 214-215 but is not mentioned in this paragraph.

RESPONSE: We added statistical test methods in the Statistical analysis section: The Chi-square test was used to compare the diagnostic performance of NTS and culture for CNS infectious diseases and the positive rate of the three detection methods (lines 187-189).

- Table 3 does not show a statistical significance comparison as mentioned in lines 195-196, it just shows the mean and median of these values. Please apply and report on the appropriate statistical testing for these parameters. The table should include detail on whether mean or median was reported and the standard deviations or interquartile ranges. The inclusion of a biostatistician may be useful.

RESPONSE: Thank you for your useful advice. We have added the necessary statistical data in Table 3, such as SD, *p*-value.

- Table 3: Abbreviations should be explained in the table footnotes.

RESPONSE: We have annotated the abbreviations in Table 3.

- Table 4: This is difficult to follow. Some lines do not have mNGS results, e.g. lines 33, 34, 41, 15. Please review and correct. It would be useful to highlight pathogens that are recognised CNS pathogens or that were thought to be significant in these patients. A clinician/clinical microbiologist could assist with this. Reporting this would help the reader determine the usefulness of these techniques in clinically suspected CNS infections and would exclude the background noise resulting from uncommon pathogens being detected by these highly sensitive techniques.

RESPONSE: Thank you for the reviewer's feedback. We have carefully reviewed Table 4 and made the necessary corrections to ensure clarity. We have also bolded the pathogens in each infectious case of Table 4, which are recognized CNS pathogens or reported to cause CNS infections or were considered significant in these patients based on the clinical symptoms. We believe that this will assist the reader in evaluating the clinical relevance of the techniques used and in distinguishing clinically relevant pathogens from background noise. We appreciate your input in improving the quality of the manuscript.

Minor comments:

- Line 52: Replace “non-specific symptoms” with “overlapping and non-specific symptoms”

RESPONSE: Thank you for your valuable feedback. We have revised it in the manuscript (line 62).

- Line 166: Please check the spelling of “Matridx” and provide the company headquarters for the company mentioned.

RESPONSE: We have checked and revised them accordingly (lines 162-164).

- In Table 2, the first line reads “age (year), median” but only a single number is given. The caption should be more descriptive.

RESPONSE: We have revised it in the Table 2.

- Line 185: Replace “meningitis stimulation” with “meningeal inflammation”

RESPONSE: We have revised it in the manuscript (line 197).

- Line 212: Correct spelling “epidermidis”.

RESPONSE: We have checked and revised them accordingly (line 223).

- Tables 5 and 6: What does the diagnostic accordance rate refer to? Please provide a definition in the methods, this is not a standard term.

RESPONSE: The diagnostic coincidence rate means the consistency between the diagnosis based on the test results of a method (NTS) and the final diagnosis results. We have added this definition to the notation in Table 5.

- In Table 5, suggest leaving out the Chi-square value and just reporting the p-value.

RESPONSE: We have revised it in the Table 5.

- Lines 234-235: Can the authors explain why specimen 23 was totally discrepant between culture and NTS? Could there have been laboratory contamination with the coagulase negative staphylococcus during culture? Could the mNGS help to clarify the discordance?

RESPONSE: Thank you for your insightful comments regarding specimen 23. We acknowledged the discrepancy between the culture and NTS results for this specimen. The CSF culture only shows the presence of viable bacteria. The empirical usage of antibiotics can lead to negative culture results. It’s possible that laboratory contamination with the coagulase negative staphylococcus during culture might account for this situation. NTS and NGS could detect the nucleic acid of pathogens which were not affected by antibiotic treatment. We agree that mNGS could potentially help to clarify the discordance observed in specimen 23. We will consider incorporating mNGS analysis to provide a more comprehensive understanding of the microbial composition in future studies.

- Line 256: The sentence cannot start with a number. Please write out “Seventy percent” or reword the sentence.

RESPONSE: We have revised it in the manuscript (line 268-269).

- Line 283: Reword to “samples exposed to antibiotics”.

RESPONSE: We have revised it in the manuscript (line 293).

Reviewer #1 (Comments for the Author):

The article titled "A Prospective Study of Nanopore Targeted Sequencing in the diagnosis for Central Nervous System Infections" by Fu, Y et. al., presents the possibility of nanopore sequencing as an alternative to diagnose neurological infections. The article highlights the existing difficulty in the diagnosis of neurological infections effectively and provides a comprehensive comparison of the conventional gold-standard methods such as culture against PCR and latest metagenomic and nanopore sequencing technologies. It shows that the gold-standard culture is not always the best method considering the variety of factors involved including the microbial load and ability of the microbe to grow in culture. The work effectively emphasizes on the need for better diagnostic tools and provide an alternative. The authors perspective and discussion on the effect of the total cellular content in the CSF is interesting. It is a commendable work that makes an important contribution to the field.

RESPONSE: Thank you very much for your recognition.

However, there are a few concerns regarding the work that I feel need to be addressed before considering for the publication.

Major concerns:

1. The authors have considered 50 CSF samples from suspected neuro-infection cases and considered the ones that were clinically exempted as non-infection as negative controls. However, the manuscript failed to inform the readers the basis of this clinical outcome.

RESPONSE: Thank you for your instructive suggestion. We mentioned in the introduction that the final diagnosis of the 50 patients in this study were the diagnosis made when discharge, based on the comprehensive consideration of patients' clinical manifestations, laboratory tests results, and patients' prognosis after antibiotic treatment (lines 120-123). Can this paragraph be used to explain what the clinical outcomes of all patients are based on.

2. Though the authors have included a no template control for illumine sequencing as a control, there is no mention of a similar control for the Nanopore sequencing.

RESPONSE: Thanks for this constructive suggestion. The same as mNGS, the control for NTS is also Tris-EDTA buffers (no-template control, NTC). We have added the description to the manuscript (lines 143-144).

In addition, the reviewer think that additional true negative controls is necessary to substantiate the suitability of the method for diagnostic purposes. Probably a CSF samples collected from patients diagnosed with non-neuroinfectious condition would serve as better controls to calculate the false negative rate of the sequencing methods

RESPONSE: Thanks for this constructive suggestion. We agree with the reviewers that additional true-negative controls are necessary, but CSF samples from patients diagnosed with non-neuroinfectious diseases are difficult to collect, and limiting factors often include clinician orders as well as patient wishes. We will focus on the collection of true-negative control samples in the future.

3. It would be interesting to see how additional positive controls of a few of the microbial species, perhaps from pure cultures would perform through the same sequencing pipeline. This would provide better statistics for the use of this method as diagnostic tool.

RESPONSE: Thank you for your instructive suggestion. In fact, the evaluation of performance by NTS was based on the standard strains and mock community as the positive controls. The standard strains were purchased from the American Type Culture Collection (ATCC) and could be separately identified correctly by NTS at the species level. Mock communities were composed of three bacteria and three fungi (*Moraxella catarrhalis*, *Acinetobacter baumannii*, *Staphylococcus aureus*, *Candida glabrata*, *Candida parapsilosis*, *Candida tropicalis*). They were constructed to reproduce a complex clinical specimen, and the proportion of each strain was changed and determined by calibration curves from cultures. Detailed steps for NTS analysis were described in our previous study (ref. 17).

Minor Comments:

1. It would help to specify the criteria for "low-quality reads" and the "undesired length"

RESPONSE : Thank you, we have added the criteria for "low-quality reads" and the "undesired length"(lines 147-148).

2. The results of the MALDI-TOF experiments need to be included in the Table 4

RESPONSE : Thank you for your insightful comments. We mentioned in the Microbial Culture that CSF sample was collected and inoculated into chocolate and blood agar plates and incubated at 35 °C for 12-72h. The final identification and characterization of pathogens was done by mass spectrometry (MS) (VITEK MS system, bioMerieux, France). VITEK MS uses MALDI-TOF (Matrix-Assisted Laser Desorption Ionization - Time of Flight) technology to clearly and accurately identify the species, genus, or group class of a sample. This means that the culture results in Table 4 are the results of MALDI-TOF experiments.

3. It would help to provide the basis and the final clinical diagnosis of these patients

RESPONSE : Thanks for this constructive suggestion, the clinician comprehensively considers the patient's clinical manifestations, laboratory examination results and the patient's prognosis after antibiotic treatment to make the final diagnosis (lines 120-123), which is shown in Table 4 "Clinical diagnosis".

Reviewer #2 (Comments for the Author):

In this article, Li and colleagues conducted a diagnostic efficacy assessment of Nanopore Targeted Sequencing (NTS) for its clinical application in central nervous system (CNS) infections. They compared it with current commonly used mNGS detection methods and the gold standard culture method on cerebrospinal fluid (CSF) from 50 patients with suspected CNS infection. And CSF with disputed test results were further detected by 16S rRNA

amplification and Sanger sequencing. This study represents a novel application of NTS technology in the field of CNS infections.

RESPONSE : Thank you very much for your recognition.

Comments:

1. In evaluating Nanopore's diagnostic efficacy for CNS, there may be an insufficient number of cases. In this manuscript, 50 patients with suspected CNS infection were enrolled, while only 31 of them were diagnosed with CNS infection. It could be beneficial to validate its performance for specific types of pathogens, such as assessing the sensitivity and specificity of NTS in CNS fungal infections.

RESPONSE : Thank you, the number of specimens is indeed a vexing problem and also a limitation that our article cannot get rid of at present, and we have added this limitation to the manuscript (line 328).

2. In clinical practice, mNGS has already been used in many hospitals for detecting pathogens in CSF. The authors could further compare NTS and mNGS detection in terms of reporting time and expenses to stress the practical value of application of NTS in CNS infection.

RESPONSE : Thank you for your instructive suggestion. We describe the reporting time of the three methods in the text (lines 321-323). However, regarding the expenses of mNGS and NTS, while costs have been greatly reduced in generating sequence data, the cost of sample reagents for sequencing is still quite high, and most laboratories lack machinery, resulting in a total cost of hundreds to thousands of dollars per sample analysis, higher than many other clinical tests. So we did not describe in the text that NTS has a good cost advantage.

3. Overall, the authors should further emphasize the application value and advantages of NTS in CNS infections.

RESPONSE: Thanks for this constructive suggestion. We agree with the reviewer's view and emphasize the application value and advantages of NTS in CNS infection in the introduction, discussion and conclusion.

Additionally, it would be beneficial to include images representing the application of NTS technology in CNS infection for better understanding.

RESPONSE: Thank you for your insightful comments. In fact, the clinical application of NTS is not widely used in this region, so there is no good image to describe the application of NTS technology in CNS infection at present. However, once CNS infection is suspected, CSF examination of patients can be performed according to the process of Figure 1 combined with multiple detection methods, so as to quickly identify the pathogen and guide clinicians in the early use of drugs to control disease progression and reduce patients' suffering.

Re: Spectrum03317-23R1 (A Prospective Study of Nanopore Targeted Sequencing in the diagnosis for Central Nervous System Infections)

Dear Dr. Yirong Li:

Thank you for the privilege of reviewing your work. Below you will find my comments, instructions from the Spectrum editorial office, and the reviewer comments.

Thank you to the authors for submitting their revised manuscript. There are still a few areas to be addressed. The comments given indicate principles that should be followed throughout the manuscript where appropriate, and are not restricted to the exact example outlined.

Major comments:

- An English language review was recommended in the comments on the original manuscript. It is not clear whether the authors have done this as there are no highlighted edits along these lines in the revised manuscript, and the manuscript still retains grammatical errors throughout. Please address.
- Statistical analysis: The authors do not report the use of a t-test, but this statistical test is quoted in Table 3. Please ensure that the analysis paragraph reflects what was done. 95% confidence intervals are needed for context of the diagnostic accordance rate in Table 6.
- Table 4 remains difficult to follow. Please include the reason for certain organisms being displayed in bold, as a footnote to the table. Please also define what neg and pos mean with regard to mNGS. What does a negative validation? Is it that mNGS was positive but had a different species to what NTS showed? It is not easy to follow.
- Table 4's results suggest that if NTS and culture were discordant, mNGS was done and 16S rDNA was only performed in cases where mNGS and NTS were discrepant. Is this correct? This does not match what is stated in the text in lines 103-104: If NTS results were inconsistent with the culture or mNGS results, 16S rDNA amplification were then carried out followed by Sanger sequencing. (but Table 4 shows that 16S did not always occur when NTS and culture were discordant, as in Samples 6, 9, 14 and others in Table 4). Please correct where necessary to make sure that this is consistent for the reader. This also applies to the abstract.
- Results, lines 222-224: It is not clear how the authors arrived at a denominator of 88 strains with mNGS. This needs to be explained.

Minor comments:

- In all places in the manuscript, please do not start a sentence with a number, e.g. line 256. This should be written out in words.
- Line 126: Change to "patient's clinical response to antibiotic treatment".
- Please be consistent when referring to diagnostic performance. The authors use "diagnostic efficiency" in line 106 but "diagnostic efficacy" in line 261.

Revision Guidelines

Data availability: ASM policy requires that data be available to the public upon online posting of the article, so please verify all

links to sequence records, if present, and make sure that each number retrieves the full record of the data. If a new accession number is not linked or a link is broken, provide Spectrum production staff with the correct URL for the record. If the accession numbers for new data are not publicly accessible before the expected online posting of the article, publication may be delayed; please contact production staff (Spectrum@asmusa.org) immediately with the expected release date.

Sincerely,
Kessendri Reddy
Editor
Microbiology Spectrum

Dear Editor and Reviewers:

Thank you for your re-evaluation of our manuscript, according to their suggestions, we have carefully revised our paper and solved all their problems. **We have marked both the previous and current review comments in the "Marked Up Manuscript File"**. Revised portions are marked in red in the paper and point-by-point responses to these comments are listed below.

Best regards,

Yirong Li

Liyirong838@163.com

Response letter to Editor and Reviewers

Manuscript ID number: Spectrum03317-23

Title of paper: A Prospective Study of Nanopore Targeted Sequencing in the diagnosis for Central Nervous System Infections

Major comments:

- An English language review was recommended in the comments on the original manuscript. It is not clear whether the authors have done this as there are no highlighted edits along these lines in the revised manuscript, and the manuscript still retains grammatical errors throughout. Please address.

RESPONSE: Thank you for your suggestion. According to the instructions of the journal, we have uploaded a version of the manuscript named "Marked-Up Manuscript". You can see the changes we made to the original manuscript based on the reviewer's comments (marked in red). Another version of the manuscript called "Revised Manuscript" has all marks removed and is a clean revised manuscript.

- Statistical analysis: The authors do not report the use of a t-test, but this statistical test is quoted in Table 3. Please ensure that the analysis paragraph reflects what was done.

RESPONSE: I apologize for our mistake. And we have added t-test to the Statistical analysis section (lines 186-188).

95% confidence intervals are needed for context of the diagnostic accordance rate in Table 6.

RESPONSE: Thank you, we have added 95% confidence intervals for each diagnostic

accordance rate in Table 6.

- Table 4 remains difficult to follow. Please include the reason for certain organisms being displayed in bold, as a footnote to the table. Please also define what neg and pos mean with regard to mNGS. What does a negative validation? Is it that mNGS was positive but had a different species to what NTS showed? It is not easy to follow.

RESPONSE: Thank you for your valuable feedback. Bacterial names in bold highlight the pathogen diagnosed as CNS infections. And "neg" in mNGS means that the mNGS results were not consistent with the species shown by the NTS, and may detect bacteria or may not detect anything at all, "pos" in mNGS means the results of mNGS were consistent with those of NTS. We have revised the footnotes in Table 6.

- Table 4's results suggest that if NTS and culture were discordant, mNGS was done and 16S rDNA was only performed in cases where mNGS and NTS were discrepant. Is this correct? This does not match what is stated in the text in lines 103-104: If NTS results were inconsistent with the culture or mNGS results, 16S rDNA amplification were then carried out followed by Sanger sequencing. (but Table 4 shows that 16S did not always occur when NTS and culture were discordant, as in Samples 6, 9, 14 and others in Table 4). Please correct where necessary to make sure that this is consistent for the reader. This also applies to the abstract.

RESPONSE: The reviewers are right, if NTS and mNGS were discordant, 16S rDNA was performed in cases. Because the results of the NTS and mNGS tests were obtained quickly, while the results of traditional cultures took longer, once the NTS and mNGS results were found to be inconsistent, we performed 16S rDNA and then Sanger sequencing. We have revised the expression in the manuscript (lines 30-31, 97, 488).

- Results, lines 222-224: It is not clear how the authors arrived at a denominator of 88 strains with mNGS. This needs to be explained.

RESPONSE: Thank you for your valuable comments. We mentioned in the manuscript that a total of 88 bacterial strains belonging to 27 species were detected by mNGS (line 214). Although we don't show the exact results in the Table 4, the mNGS column in Table 4 shows "neg" and "pos". As answered in comment 3: the "neg" in mNGS means that the mNGS results were not consistent with the species shown by the NTS, and may detect bacteria or may not detect anything at all, "pos" in mNGS means the results of mNGS were consistent with those of NTS. What we describe later is the detection rate of a certain bacteria by mNGS, so 88 is the total number of detections as the denominator, and the detection frequency of a certain bacteria as the numerator.

Minor comments:

- In all places in the manuscript, please do not start a sentence with a number, e.g. line 256. This should be written out in words.

RESPONSE: Thank you for your kind instructions. We have revised it in the manuscript (lines 27, 131, 192, 196, 209)

- Line 126: Change to "patient's clinical response to antibiotic treatment".

RESPONSE: We have revised it in the manuscript (lines 119-120)

- Please be consistent when referring to diagnostic performance. The authors use "diagnostic efficiency" in line 106 but "diagnostic efficacy" in line 261.

RESPONSE: Thank you, we changed line 254 to "diagnostic efficiency".

Re: Spectrum03317-23R2 (A Prospective Study of Nanopore Targeted Sequencing in the diagnosis for Central Nervous System Infections)

Dear Dr. Yirong Li:

Your manuscript has been accepted, and I am forwarding it to the ASM production staff for publication. Your paper will first be checked to make sure all elements meet the technical requirements. ASM staff will contact you if anything needs to be revised before copyediting and production can begin. Otherwise, you will be notified when your proofs are ready to be viewed.

Sincerely,
Kessendri Reddy
Editor
Microbiology Spectrum